# Walking with a Mobile Phone: A Randomised Controlled Trial of Effects on Mood

Randi Collin and Elizabeth Broadbent *

Department of Psychological Medicine, The University of Auckland, Auckland 1142, New Zealand
* Correspondence: e.broadbent@auckland.ac.nz

**Abstract:** It is now common to see pedestrians looking at their mobile phones while they are walking. Looking at a mobile phone can cause stooped posture, slower gait, and lack of attention to surroundings. Because these walking characteristics have been associated with negative affect, walking while looking at a mobile phone may have negative effects on mood. This study aimed to investigate whether walking while looking at a mobile phone had psychological effects. One hundred and twenty-five adults were randomised to walk in a park either with or without reading text on a mobile phone. Participants wore a fitness tracker to record pace and heart rate, and posture was calculated from video. Self-reported mood, affect, feelings of power, comfort, and connectedness with nature were assessed. The phone group walked significantly slower, with a more stooped posture, slower heart rate, and felt less comfortable than the phone-free group. The phone group experienced significant decreases in positive mood, affect, power, and connectedness with nature, as well as increases in negative mood, whereas the phone-free group experienced the opposite. There was no significant mediation effect of posture on mood; however, feeling connected with nature significantly mediated the effects of phone walking on mood. In conclusion, individuals experience better wellbeing when they pay attention to the environment rather than their phone while walking. More research is needed to investigate the effects of performing other activities on a mobile phone on mood while walking and in other settings.

**Keywords:** mobile phone; mood; nature; walk; attention





## 1. Introduction

Looking at a mobile phone while walking (phone walking) is relatively common. One in four pedestrians at New York intersections exhibited distracted walking from mobile devices [1]. Similarly, in the UK, 31% of pedestrians had a mobile device observable when crossing the road [2]. Furthermore, a meta-analysis of eight observational studies found that the percentage of pedestrians distracted by a phone when crossing a road ranged from 12% to 45% [3].

Phone walking requires a split of attention, which favours the phone task [4]. Pedestrians' awareness of stimuli in the immediate environment becomes markedly impaired when using a phone and can result in accidents [5]. Experimental studies have shown that talking, texting, and browsing on a mobile phone reduces the frequency of looking left or right when crossing a road and increases collisions and near misses with vehicles [3].

Walking with a mobile phone also compromises postural stability, with significant head, neck, and thorax flexion, reduced vertical sway movements, and increased lateral head flexion [6]. People walk more slowly and deviate more from a straight line when looking at a smart phone [7]. Texting while walking also reduces walking speed and shortens stride length [8].

The gait and postural characteristics associated with phone walking overlap considerably with those associated with negative affect. Major depression can be characterised by a stooped posture, slower and smaller steps, reduced head movement, and swaying upper

body movements [9]. A systematic review found consistent evidence that depression is associated with slower walking, reduced stride length, and abnormal posture and sway movement [10].

It is possible that the stooped posture and impaired gait that result from using a mobile phone while walking could influence mood states. This is because, according to the theory of embodied cognition, physiological changes in muscle tone, posture, movement, and facial expressions are linked to specific emotions and cognitions [11]. Research supports this theory, showing that sitting or walking with a slumped posture can cause negative affective states, lower feelings of power, and lower physiological arousal compared to upright posture [12–15]. Walking in a 'sad' style has also been shown to cause negative bias in processing emotion-related information [16]. These results are further supported by a meta-analysis of 88 studies, which found significant and consequential effects of body positions on self-reported feelings, as well as behaviours [17].

To date, no research has examined whether the specific postural and gait patterns associated with phone walking can affect mood. This study aimed to explore the psychological effects of phone walking. We hypothesised that phone walking would cause stooped posture, slower walking, lower arousal, and worse mood and affect than walking without a phone. The primary outcome was overall mood. For safety reasons, walking was performed in a park on a grass track with surrounding mature trees.

It is important to note that attention to a phone rather than to the surrounding natural environment may also have effects on mood. More time spent on a smartphone has been associated with lower connectedness with nature [18]. Furthermore, walking in natural environments has been shown to have additional benefits for wellbeing compared to walking in synthetic environments [19,20], and natural environments have been shown to have large effects on positive affect compared to urban environments [21]. Therefore, a further hypothesis was that phone walking would reduce attention to the park environment compared to walking without a phone, as indexed by feeling less connected to nature.

## 2. Methods

### Participants

The study was advertised on university email lists. The inclusion criteria were: (1) being able to understand and read English and (2) being aged 16 years or older. Exclusion criteria were: (1) impaired vision that prohibited individuals from reading text on a smart phone, (2) unable to walk unaided for one kilometre (approximately 12 min).

A total of 157 people responded to the advertisement and underwent eligibility assessment. Of the 32 people excluded, 2 declined to participate, 20 either did not respond to a further invitation to set a time to participate or could not find a suitable time to participate, and 10 were excluded due to recruitment being complete (see Figure 1). Of the 125 participants, one was excluded due to not following the walking track correctly. The final sample consisted of 124 participants, of whom 84 (68%) were female. Participants ranged in age from 17 to 86 years ($M = 26.05$, $SD = 10.57$). The majority were either Asian ($n = 65$, 52%) or New Zealand European ($n = 58$, 47%), single ($n = 97$, 78%), and university students ($n = 115$, 93%).

A power analysis based on a medium effect size (partial $\eta^2 = 0.06$) [12], power of 0.80, and alpha of 0.05, indicated that a total sample size 125 was required.

The study was approved by the University of Auckland Human Participants Ethics Committee on 24 March 2017. The study was carried out in accordance with the provisions of the World Medical Association Declaration of Helsinki. It was pre-registered with the Australian New Zealand Clinical Trials Registry on 10 May 2017 (Trial ID: AC-TRN12617000674336). All participants provided written informed consent.

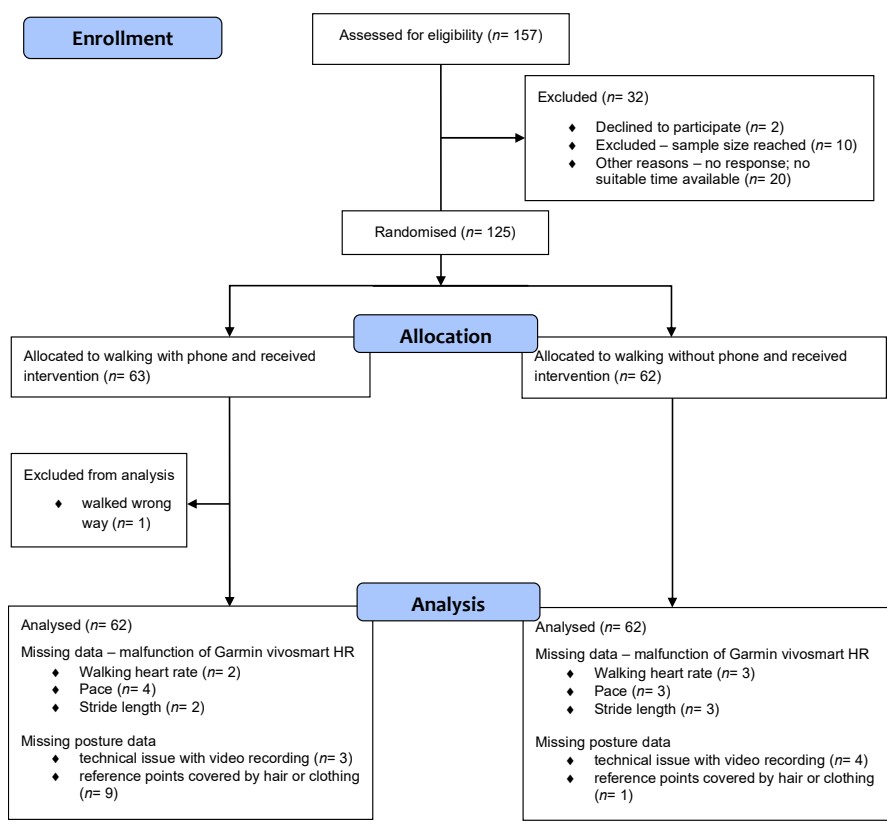

**Figure 1.** Consort Flow Chart.

### 3. Measures

Three Visual Analogue Scales (VAS) were administered before and after walking to assess current overall mood, positive mood, and negative mood. Participants rated how they felt right now by putting an "x" on a 100 mm long horizontal line. The anchor labels at the ends of each line were '*worst*' to '*best*', '*not positive at all*' to '*very positive*', and '*not negative at all*' to '*very negative*' for each scale, respectively.

A modified version of the Affect Valuation Index (AVI) [22] was administered before and after walking to assess actual present affective states. Each item offered five response options ranging from 1 (*very slightly* or *not at all*) to 5 (*extremely*). The items belong to eight different subscales: high arousal positive (enthusiastic, excited, strong, elated); positive (content, happy, satisfied); low arousal positive (rested, calm, peaceful, relaxed, serene); low arousal negative (dull, sleepy sluggish); negative (lonely, sad, unhappy); high arousal negative (fearful, hostile, nervous); and high arousal (aroused, astonished, surprised). Internal consistency ranged from 0.58 to 0.87 for these subscales, and test-retest reliability was 0.65 [22]. This version has previously been used in postural research [13].

A Feelings of Power questionnaire [23] was administered before and after walking. It assessed how dominant, in control, and powerful participants felt on a 1 (*not at all*) to 5 (*extremely*) scale with Cronbach's alpha of 0.82. An additional item assessing how confident participants felt was added to make a four-item scale; Cronbach's alpha for this scale at baseline was also 0.82.

The Inclusion of Nature in Self Scale (INS Scale) [24] was used to measure connectedness with nature. Participants' general connectedness with nature (trait) was only assessed at baseline, while current levels of connectedness with nature (state) were assessed pre and post walking. At each time-point participants were asked to choose one of seven different response options showing two circles, labelled self and nature, with different amounts of overlap ranging from complete overlap to total separation, representing various levels of connectedness. As this is a one-item measure, scale reliability indices cannot be calculated.

Degree of physical comfort was measured on a visual analog scale before and after walking. The instructions asked participants to indicate their current level of physical comfort with an "X" on a 100 mm horizontal line with the anchor labels, *not comfortable at all* and *very comfortable*, beside the ends of the line.

Blood pressure and heart rate were obtained from an automatic sphygmomanometer (Scian LD-582) after completing the baseline questionnaire but before walking, as well as straight after completing the walking task. To ensure reliable readings, the cuff was applied to the left upper arm at heart level, either with all arm clothing removed or over very thin material only. Participants were seated during the procedure, feet flat on the ground, and with the left arm placed on their left leg to avoid disruptive movement during the reading.

Gait characteristics including walking speed, step count, stride length, and heart rate during walking were obtained by the Garmin vivosmart HR wrist-worn fitness-tracker. This was fastened securely slightly above the left wrist.

Head and neck posture were determined based on still frames taken from videos of each participant walking. The video camera (Panasonic HX-DC2) was positioned 1.6 m off the ground on a tripod with an approximate lateral distance of 4.5 to 5 m to the walking route. Still frames were printed, and head and neck flexion angles determined manually, following the procedure described by Guan et al., [25]. Head flexion angles were measured using an acute angle with the vertex on the tragus, one side of the angle at the canthus, and the other side of the angle parallel to the vertical plane. Neck flexion angles were measured using an acute angle with the vertex on the C7 spinous process, one side of the angle on the tragus and the other side of the angle parallel to the vertical plane.

## 4. Procedure

A one kilometre walking track was set around the perimeter of grass sports (cricket) fields in a park (see Figure 2). The season was winter, so there were no games played during the study. Upon arrival, each participant was informed about the study procedure, offering a broad explanation about the study's purpose to assess the health effects of phone walking, not mentioning specific interest in posture, gait, and affective states. Participants signed a consent form and completed the baseline questionnaire. Afterwards, the first blood pressure measure was taken, and the fitness tracker was fitted. Next, participants received verbal and visual instructions about the walking route before they were randomly allocated to one of the two experimental conditions.

Participants in the phone group were instructed to read the informative text about the park on the smart phone (5.1-inch Samsung GalaxyS5) as best as possible, while keeping glances away from the phone to a minimum. Participants in the phone-free group were instructed to pay attention to their environment. Both were told that there would be a short test at the end about the park, to make sure they followed the instructions; the test had no relevance to the study and was not analysed.

Then, the fitness tracker was set up, started, and each participant began their walk. Part-way through the walk, participants walked past the video camera, which provided a side-on recording of the upper body. After completing the walk, a second blood pressure measure was taken, and participants filled in the post-walking questionnaire. This included the same measures as baseline.

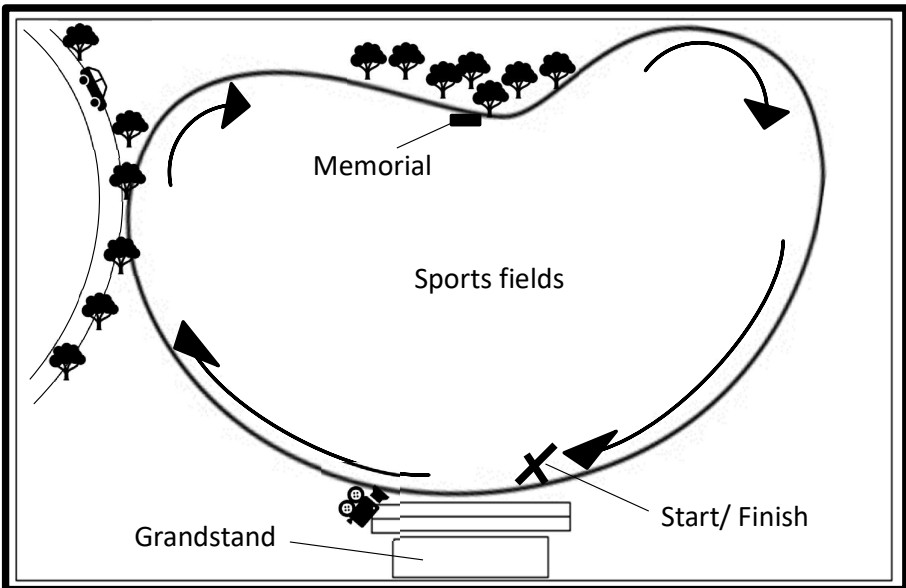

**Figure 2.** Map of the walking track given to participants showing start and finish points. The direction of the walk around the sports fields is indicated by the arrows. The camera location is at the left of the grandstand.

## 5. Data Analysis

All data were analysed using IBM SPSS 23. Independent *t*-tests and Pearson's chi-square tests were applied to check data for group differences at baseline. Analyses of covariance (ANCOVAs) were conducted to investigate whether change scores between pre- and post-walking for affect, mood, power, connectedness with nature, comfort, and blood pressure differed between groups after controlling for baseline scores. This analysis helps take into account any differences at baseline and regression to the mean. Independent-samples *t*-tests were used to compare scores between groups in head and neck flexion angles, attention, and average heart rate while walking.

Mediation analyses were conducted using the PROCESS custom dialogue box for SPSS [26] to determine the influence of posture, arousal, and connectedness with nature on the relationship between phone-walking and overall mood after walking.

Neither the data nor the materials have been made available on a permanent third-party archive; requests for the data or materials can be sent via email to the corresponding author.

## 6. Results

*6.1. Baseline Differences*

There were no significant baseline differences between the experimental groups for any variables with one exception. The phone-free group reported significantly greater low arousal affect before walking (*M* = 12.44, *SD* 3.95) than the phone group (*M* = 10.31, *SD* 3.15), *t* (122) = −3.32, *p* = 0.001.

*6.2. Mood, Affect, Power, and Connectedness with Nature*

There were significant differences between the phone and phone-free groups in changes to self-reported mood, affect, power, comfort, and connectedness with nature after walking, as shown in Table 1.

**Table 1.** Mean changes in mood, affect, power, state connectedness with nature, comfort, and blood pressure in each walking group assessed by ANCOVA controlling for baseline scores.

| Variable | Phone (*n* = 62) $M_{adj}$ (SD) | Phone-Free (*n* = 62) $M_{adj}$ (SD) | F | df | p | $\eta_p^2$ | 95% CI |
|---|---|---|---|---|---|---|---|
| **Mood** | | | | | | | |
| Positive mood | −5.21 (15.55) | 6.00 (15.55) | 15.98 | 1, 121 | **0.000** | 0.12 | [−16.76, −5.66] |
| Negative mood | 5.99 (16.51) | −5.18 (16.51) | 14.07 | 1, 121 | **0.000** | 0.10 | [5.28, 17.07] |
| Overall mood | −1.89 (12.98) | 7.83 (12.98) | 17.29 | 1, 121 | **0.000** | 0.13 | [−14.35, −5.09] |
| **Affect** | | | | | | | |
| High arousal positive affect | −1.53 (2.81) | 0.44 (2.81) | 14.20 | 1, 121 | **0.000** | 0.11 | [−2.97, −0.97] |
| Positive affect | −1.03 (1.76) | 0.40 (1.76) | 20.63 | 1, 121 | **0.000** | 0.15 | [−2.06, −0.81] |
| Low arousal positive affect | −2.24 (3.43) | 0.50 (3.43) | 19,56 | 1, 121 | **0.000** | 0.14 | [−3.97, −1.52] |
| Low arousal affect | −1.08 (3.03) | −2.28 (3.03) | 4.69 | 1, 121 | **0.032** | 0.04 | [0.10, 2.31] |
| Low arousal negative affect | −0.21 (1.88) | −1.19 (1.88) | 8.41 | 1, 121 | **0.004** | 0.07 | [0.31, 1.65] |
| Negative affect | 0.17 (1.06) | −0.10 (1.06) | 2.05 | 1, 121 | 0.155 | 0.02 | [−0.10, 0.65] |
| High arousal negative affect | −0.25 (1.02) | −0.35 (1.02) | 0.31 | 1, 121 | 0.577 | 0.00 | [−0.26, 0.47] |
| High arousal affect | 0.52 (1.28) | 0.80 (1.28) | 1.44 | 1, 121 | 0.233 | 0.01 | [−0.73, 0.18] |
| **Power** | −0.97 (2.35) | 0.60 (2.35) | 13.73 | 1, 121 | **0.000** | 0.10 | [−2.41, −0.73] |
| **Connectedness to nature** | −0.35 (1.08) | 0.89 (1.08) | 40.82 | 1, 121 | **0.000** | 0.25 | [−1.63, −0.86] |
| **Physical comfort** | −1.38 (13.83) | 7.31 (13.83) | 12.20 | 1, 121 | **0.001** | 0.09 | [−13.61, −3.76] |
| **Systolic blood pressure** | 5.26 (12.19) | 9.40 (12.19) | 3.58 | 1, 121 | 0.061 | 0.03 | [−8.48, 0.19] |
| **Diastolic blood pressure** | 0.34 (6.89) | 1.42 (6.89) | 0.77 | 1, 121 | 0.383 | 0.01 | [−3.53, 1.37] |

*Note.* *n* = sample size; $M_{adj}$ = adjusted means; SD = standard deviation of $M_{adj}$; F = F-statistic; *df* = degrees of freedom; *p* = *p*-value; $\eta p^2$ = partial eta squared; CI = confidence interval [upper, lower bound]. Bolded *p* indicates statistical significance (*p* < 0.05).

Phone walking led to a worsening of all visual analog mood measures (positive, negative, and overall), whereas walking phone-free improved all these mood measures. Similarly, phone walking reduced all three positive affect subscales of the AVI, whereas phone-free walking led to improvements. Low arousal affect and low arousal negative affect decreased in both groups, but significantly more in the phone-free group. There were no significant differences between groups in changes to negative arousal, high arousal negative affect, or high arousal after walking.

Phone-walking lowered feelings of power, while phone-free walking increased feelings of power. Changes in state connectedness with nature were also significantly different between groups. Phone-walking resulted in a drop of connectedness with nature, whereas phone-free walking increased connectedness with nature. Physical comfort was significantly higher in the phone-free group.

*6.3. Posture, Gait, and Physiology*

Phone-walking resulted in a significantly greater head flexion angle (*M* = 103.76, *SD* 10.56) than walking phone-free (*M* = 83.77, *SD* 12.53), *t* (105) = 8.85, *p* < 0.001, *d* = 1.89, mean difference 95% CI [15.20, 24.42]. Phone walking also increased neck flexion angle (*M* = 56.89, *SD* 7.10) relative to walking phone-free (*M* = 46.24, *SD* 8.84), *t* (105), = 6.81, *p* < 0.001, *d* = 1.50, mean difference 95% CI [7.59, 13.52].

The walking pace was significantly slower in the phone group (*M* = 4.63 km/h, *SD* 0.42) than the phone-free group (*M* = 4.87 km/h, *SD* 0.47), *t* (117) = −2.96, *p* = 0.003, *d* = 0.51, mean difference 95% CI [−0.39, −0.08]. Stride length did not significantly differ between groups, *t* (117) = −0.53, *p* = 0.618, *d* = 0.01., mean difference 95% CI [−0.02, 0.01].

Average heart rate when walking was significantly slower in the phone group (*M* = 106.43, *SD* 11.55) than the phone-free group (*M* = 111.81, *SD* 14.72), *t* (117) = −2.22, *p* = 0.027, *d* = 0.47, mean difference 95% CI [−10.31, −0.076]. There were no significant differences in the changes in systolic or diastolic blood pressure from baseline to after the walking task between groups (Table 1).

*6.4. Mediation Analyses*

Change in overall mood after walking was significantly correlated with neck flexion angle ($r = -0.22$, $p = 0.021$) and change in connectedness with nature after walking ($r = 0.43$, $p < 0.001$), but not walking pace ($r = -0.15$, $p = 0.108$), head flexion angle ($r = -0.11$, $p = 0.241$), or heart rate ($r = 0.02$, $p = 0.819$). Mediation analysis showed no significant indirect effects of neck angle on change in overall mood after walking (controlling for baseline overall mood), $ab = 0.63$, BCa CI [$-2.20$, 3.84]. However, there was a significant indirect effect of change in connectedness with nature on overall mood after walking (after controlling for baseline mood), $ab = 4.54$, 95% BCa CI [1.74, 7.61], Figure 3. The mediator could account for about half the effect of phone walking on change in mood, $P_M = 47\%$.

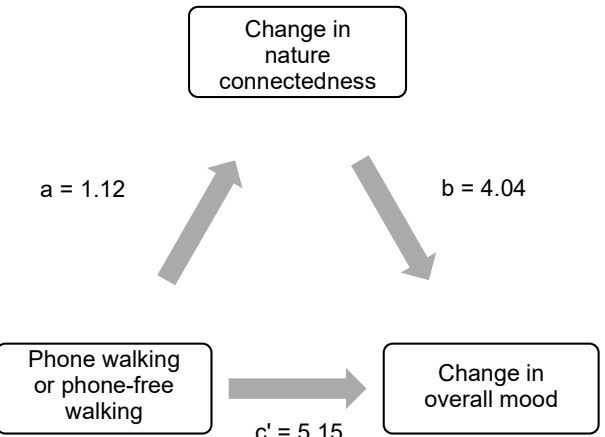

**Figure 3.** Mediation model for the effects of phone walking on mood via change in connectedness with nature.

## 7. Discussion

This is the first study to assess the effects of phone walking on mood. The results support the hypotheses that phone walking leads to a more stooped posture, slower walking, slower heart rate, more negative and less positive affective states, as well as lower feelings of power, compared to walking phone-free. These results suggest that walking with a mobile phone reduces the positive effects of exercise on mood that have previously been demonstrated [27,28].

The findings support previous research showing that texting while walking reduces walking speed [8], although stride length was not significantly affected, possibly because participants were reading rather than texting on the phone. The results also align with research showing that browsing on a mobile phone while walking affects head flexion angle [29].

The negative effects of phone walking on mood may be due to a bent neck angle. Stooped posture has been shown to increase low arousal negative affect and alertness compared to upright posture while walking [12,30]. In this study, although a bent neck angle was associated with worse overall mood, the mediation analysis did not support this explanation.

An alternative explanation for the effects of phone walking on mood is a lack of attention to the surroundings and subsequent feelings of disconnection from nature. Previous work has shown that disconnection from nature can increase feelings of depression, anxiety, and frustration [31]. Our results showed that the phone group did experience significantly lower connectedness with nature than the phone free group, which agrees with previous findings that more time spent on a phone is linked to lower nature connectedness (Richardson, 2018) [18]. In this study, connectedness to nature partially mediated the effects of phone walking on overall mood, which supports this explanation. However, this needs to be treated with caution because the measures were administered at the same timepoints, and common method bias may have artificially inflated the association.

This study has several limitations. The study was conducted in a park and results may not generalize to urban settings. Furthermore, the study only provided information on the short-term effects of phone walking, and subsequent studies could investigate longer term effects. The reading material on the phone was about the park and was neutral, and future research could investigate the effects of other phone activities on mood while walking, such as texting. The observed effects might also occur if walking while performing other similar distracting tasks, such as reading a book. However, this needs to be tested. The sample consisted mainly of university students, so the findings may not generalize to older adults, and older adults may be less likely to look at their phones while they are walking due to differences in culture and agility. Future research could also investigate whether expectations play a role.

Overall, this study suggests that reading text on a mobile phone while walking negatively affects mood compared to walking without a phone. However, these findings are preliminary, and more research is needed to investigate the effects of walking on mood while performing other phone-related tasks, as well as in other settings.

**Author Contributions:** Both authors, E.B. and R.C., contributed to study design and writing the paper. R.C. collected the data and entered and analysed the data in SPSS. All authors have read and agreed to the published version of the manuscript.

**Funding:** This research received no external funding.

**Institutional Review Board Statement:** This study was performed according to the APA ethical code. This study obtained ethics approval from the University of Auckland Human participants Ethics Committee (018789). This research was conducted following the principles within the Helsinki Declaration to ensure participants are respected, their health is safeguarded, and that they retain the right to make informed decisions throughout the study.

**Informed Consent Statement:** Informed consent was obtained from all subjects involved in the study.

**Data Availability Statement:** Neither the data nor the materials have been made available on a permanent third-party archive; requests for the data or materials can be sent via email to the corresponding author.

**Conflicts of Interest:** The authors declare no conflict of interest with respect to the authorship or the publication of this article.

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
