# Peer review of "Walking with a Mobile Phone: A Randomised Controlled Trial of Effects on Mood"

_psych, doi:10.3390/psych5030046_

Round 1

Reviewer 1 Report

1. Turnitin check: 86%. It is suggested that the authors reduce this similarity level to below 20%. 

2. The measurement of the mood of the participants might be biased. How to ensure the result was not affected by the general mood of the participants on that day? Measuring the mood before and after walking could not objectively counter this since it was only 12 minutes of walking.

The manuscript could be written in a better writing style with a better flow from one paragraph to the next paragraph.

Author Response

  1. Turnitin check: 86%. It is suggested that the authors reduce this similarity level to below 20%. 

Response –  This is original research that has not previously been published so we are not sure why this is so high. The paper is based on an a unpublished masters thesis by the first author.  We did present this work in a virtual oral presentation at the International Congress of Behavioural Medicine in 2021, for which an abstract (only) was published. We have now added more recent papers to this paper that were not cited in the thesis and improved the flow of the introduction, which should reduce this similarity score. The references section usually has a very high similarity score so this should be removed before passing the revised paper through turnitin.

  1. The measurement of the mood of the participants might be biased. How to ensure the result was not affected by the general mood of the participants on that day? Measuring the mood before and after walking could not objectively counter this since it was only 12 minutes of walking.

Response- The type of statistical analysis we performed on the mood measures (ANCOVA on change scores controlling for baseline scores) controls for any differences in the general mood of the participants on that day. Although the walking period was relatively short, there were changes in mood after walking that significantly differed between groups. Because participants were randomly assigned to groups, this difference can be attributed to the independent variable (walking with or without a phone).

  1. The manuscript could be written in a better writing style with a better flow from one paragraph to the next paragraph.

Response- We have improved the flow from one paragraph to another.

Reviewer 2 Report

this is an exceptionally well done, well described, and well executed study which is very important today, given the influce of smartphones on our cultures.  The only criticism I have is that the authors need to add the reference for Baswail, Allinson, Goddard, & Pfeffer, 2019

Author Response

this is an exceptionally well done, well described, and well executed study which is very important today, given the influce of smartphones on our cultures.  The only criticism I have is that the authors need to add the reference for Baswail, Allinson, Goddard, & Pfeffer, 2019

Response- We have added this to the reference list.

Reviewer 3 Report

First of all, I would like to congratulate the authors on their choice of topic. It seems to me to be a very interesting issue that they address in their project. However, I would like to draw attention to some areas for improvement, which need to be addressed before it is possible to publish this manuscript. In particular:

- You must comply with APA 7 standards in the citation, which state, among other issues, that you must put et al. when the study has 3 or more authors from the first time they are cited, so it is advisable to review the citation style of the entire text.

- The references are not up to date. There are many of them prior to the last 5 years, despite being a current topic, so it is necessary to go deeper into the related literature, expanding the Introduction with the results of the most relevant studies in this field in the last 5 years. At least 3-5 extra articles should be mentioned, not only citing them, but also setting out their main conclusions in writing.

- The citation of the World Medical Association Declaration of Helsinki is missing.

- Reliability indicators should be added for all the scales used. If they have been previously validated, the indicator given by the original author should be used, otherwise the one given by the study data itself can be used.

- As in the introduction, the "Discussion" section could be improved. Once the 3-5 new current references are added in the introduction, the results obtained in the "Discussion" should be compared with those obtained in this study.

Author Response

- You must comply with APA 7 standards in the citation, which state, among other issues, that you must put et al. when the study has 3 or more authors from the first time they are cited, so it is advisable to review the citation style of the entire text.

Response- We have changed these citations to use et al for three or more authors. We also looked up the reference style for MDPI journals, which uses a numbered format rather than APA. If the paper is accepted we will reformat the references and citations to the correct journal format.

- The references are not up to date. There are many of them prior to the last 5 years, despite being a current topic, so it is necessary to go deeper into the related literature, expanding the Introduction with the results of the most relevant studies in this field in the last 5 years. At least 3-5 extra articles should be mentioned, not only citing them, but also setting out their main conclusions in writing.

Response- We have now cited and described findings from 7 new papers from the last 5 years in the introduction and/ or discussion.

- The citation of the World Medical Association Declaration of Helsinki is missing.

Response- We have added this declaration at the end of the paper.

- Reliability indicators should be added for all the scales used. If they have been previously validated, the indicator given by the original author should be used, otherwise the one given by the study data itself can be used.

Response- We have added more information about reliability for the scales with more than one item.

- As in the introduction, the "Discussion" section could be improved. Once the 3-5 new current references are added in the introduction, the results obtained in the "Discussion" should be compared with those obtained in this study.

Response- We added more recent citations and updated the discussion to compare these results with previous research.

We thank all the reviewers for their comments.

Round 2

Reviewer 1 Report

Turnitin check was done by excluding the references. The similarity is now 79%; however, it is still too high for a research article. It should be less than 20%.  

The article is now easier to follow.